# Rational highly dispersed ruthenium for reductive catalytic fractionation of lignocellulose

Zhenzhen Liu[1], Helong Li [1], Xueying Gao[1], Xuan Guo[2], Shuizhong Wang [1] ✉, Yunming Fang [2] ✉ & Guoyong Song [1] ✉

Producing monomeric phenols from lignin biopolymer depolymerization in a detachable and efficient manner comes under the spotlight on the fullest utilization of sustainable lignocellulosic biomass. Here, we report a low-loaded and highly dispersed Ru anchored on a chitosan-derived *N*-doped carbon catalyst (RuN/ZnO/C), which exhibits outstanding performance in the reductive catalytic fractionation of lignocellulose. Nearly theoretical maximum yields of phenolic monomers from lignin are achieved, corresponding to TON as 431 $mol_{phenols}$ $mol_{Ru}^{-1}$, 20 times higher than that from commercial Ru/C catalyst; high selectivity toward propyl end-chained guaiacol and syringol allow them to be readily purified. The RCF leave high retention of (hemi) cellulose amenable to enzymatic hydrolysis due to the successful breakdown of biomass recalcitrance. The RuN/ZnO/C catalyst shows good stability in recycling experiments as well as after a harsh hydrothermal treatment, benefiting from the coordination of Ru species with *N* atoms. Characterizations of the RuN/ZnO/C imply a transformation from Ru single atoms to nanoclusters under current reaction conditions. Time-course experiment, as well as reactivity screening of a series of lignin model compounds, offer insight into the mechanism of current RCF over RuN/ZnO/C. This work opens a new opportunity for achieving the valuable aromatic products from lignin and promoting the industrial economic feasibility of lignocellulosic biomass.

The world needs to transition away from fossil fuels and chemicals to avoid the worst impacts of climate change[1]. Lignocellulosic biomass, formed from photosynthetic $CO_2$ captured by plants, can be produced at an annual average of *ca*. 10 metric tons per hectare[2]. Such an abundant and sustainable carbon feedstock represents a prime candidate to alternate or complement fossil carbons to produce fuels and chemicals[3–5]. Generally, lignocellulosic biomass comprises three biopolymers, i.e., cellulose (30-50%), hemicellulose (20-35%), and lignin (15-30%)[3,6]. Cellulose and hemicellulose are sugar-based polymers, which are composed exclusively of glucose units and mainly of xylose units, respectively. Lignins are complex phenylpropanoid polymers,

representing the most abundant renewable aromatic carbon resource on the earth. Lignin biopolymers are biosynthesized from three phenylalanine-derived monolignols, i.e., *p*-hydroxyphenyl (H), guaiacyl (G), and syringyl (S), which are connected by β-aryl ether (β-*O*−4) as major linkages[7,8]. In the plant cell wall matrices, lignin embraces hemicellulose and cellulose, and cellulose fibers are interlaced with hemicelluloses[9] (Fig. 1). Historically, biorefinery schemes aimed to valorize (hemi)celluloses, whereas lignin was considered an inferior and inconvenient biopolymer[10]. Conventional pretreatments for the fractionation of lignocelluloses, such as Kraft and soda processes, always lead to the irreversible formation of C−C bonds at the expense

[1]Beijing Key Laboratory of Lignocellulosic Chemistry, Beijing Forestry University, Beijing 100083, China. [2]College of Chemical Engineering, Beijing University of Chemical Technology, Beijing 100029, China. ✉e-mail: szwang@bjfu.edu.cn; fangym@mail.buct.edu.cn; songg@bjfu.edu.cn

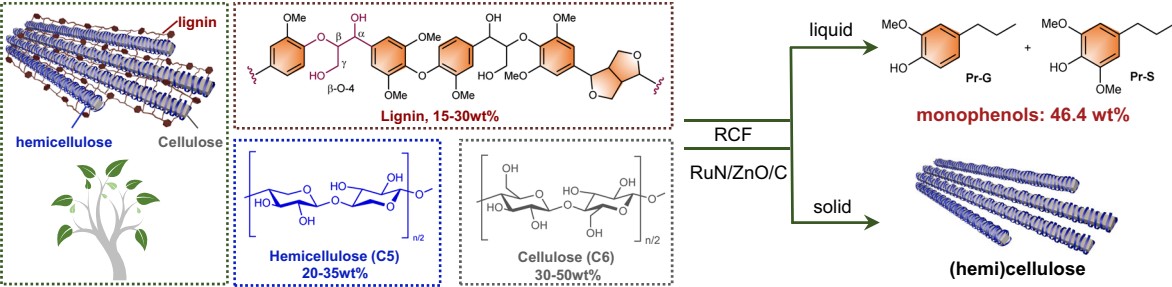

**Fig. 1 | Route for reductive catalytic fractionation of lignocellulosic biomass.** Representative structures of lignin, cellulose, and hemicellulose (left). A highly dispersed Ru catalyst (RuN/ZnO/C) is used for reductive catalytic fractionation of lignocellulose, which affords monophenols in theoretical maximum yields, as well as well-preserved (hemi)cellulose components (right).

of reactive β-*O*−4 linkages in native lignin, thus resulting in industrial lignins that are difficult to transform. Therefore, the valorization of lignin has come into the spotlight on the full utilization of lignocellulosic biomass[8,11–13].

Reductive catalytic fractionation (RCF) of biomass, which enabled the depolymerization of native lignin in first, recently became a new biorefinery paradigm for lignin valorization[14,15] (Fig. 1). The physical location and reactivity of lignin in biomass matrices allowed it to be extracted and then depolymerized into phenolic monomers, dimers, and oligomers. Furthermore, most (hemi)celluloses can be preserved in the solid phase for further transformation. The lignin-to-phenolics procedure should involve the cleavage of β-*O*−4 units and subsequent stabilization steps (such as hydrogenation and/or dehydroxylation)[15], both of which are catalyst-dependent processes. Heterogeneous catalysts based on precious metals (such as Ru[16–18], Pd[17,19–22], Pt[23,24], and Rh[23]) or nonprecious metals (such as Ni[25–29], Mo[30,31], and Co[32]) have been developed for the RCF of lignocellulosic biomass. The catalysts based on precious metals showed prominent performance compared to nonprecious metals, while the extreme scarcity and high cost of precious metals, as well as the utilization efficiency of these metals in RCF, will hardly meet industrial practicability[22]. For example, the use of commercial Ru/C and Pd/C catalysts (where Ru and Pd loading are generally 5 wt%) can lead to near theoretical lignin monomer yields (*ca.* 40-50 wt% based on the lignin component[15,33]) in the RCF of hardwoods, while the turnover numbers (TONs) were established as approximately 20 $mol_{phenols} mol_{metal}^{-1}$.

Low-loading and highly dispersed metal catalysts, which possess well-defined single atoms and/or nanoclusters metal species, have exhibited enhanced catalytic performance with low metal consumption[34–37]. These catalysts have been proven to be effective in various thermocatalysis[38], photocatalysis[39], and electrocatalysis[40,41] transformations of small molecules. Nevertheless, examples of atomically dispersed metal catalysts for the depolymerization of lignin macromolecules are still rare. Tungasmita and Samec demonstrated that the RCF of birch wood with Co-phen/C (where Co loading is 0.45 wt%) catalyst afforded phenolics with propyl and propenyl end-chains in 34 wt% yield[32]. Kim and coworkers reported a $Pd_{0.25}/CN_x$ (where Pd loading is 0.25 wt%) catalyst containing both Pd clusters and single atoms, which enabled the formation of mixed monophenols in 52.7 C% yield in the RCF of birch[22]. Laurenczy and Dyson anchored single Pt atoms onto Ni nanoparticles ($Pt_1Ni/C$, where Pt and Ni contents are 0.3 wt% and 4.4 wt%, respectively), by which monophenolic compounds were generated in 43 wt% yield during RCF of birch[24]. We also demonstrated that an atomically dispersed Ru catalyst (where Ru loading is 0.2 wt%) can efficiently catalyze the hydrogenolysis of catechyl lignin (C-lignin) into catechols through the cleavage of proximal C−O bonds in benzodioxane units[42]. Compared to Ru/C and Pd/C catalysts, RCF over the abovementioned catalysts delivered multiple monophenols tethering different end chains (propyl, propenyl, and propanol), thus resulting in separation and purification problems.

From a practical aspect, desirable metal catalysts for RCF of biomass should have the following process abilities: 1) integrating high utilization efficiency and strong stability of active metal centers, 2) producing phenolic monomers at theoretical maximum yield, together with excellent selectivity allowing for effortless purification, and 3) leaving well-preserved (hemi)cellulose components for valorization. To realize these goals, we herein design a heterogeneous catalyst comprised of spatially isolated Ru atoms stabilized by nitrogen atoms on carbon. Chitosan, a naturally occurring polysaccharide having fruitful uncoordinated amine groups[43,44], was used as the support precursor. Given that Lewis acid centers can promote the hydrogenolysis of β-*O*−4 units[45], Zn species were introduced to the catalyst. Characterizations of the as-prepared RuN/ZnO/C catalyst indicated that Ru was fabricated as single atoms on *N*-doped carbon. During RCF of biomass, Ru single atoms would be transformed into nanoclusters, the possible catalytically active species for lignin hydrogenolysis. This catalyst exhibited high activity for RCF of birch wood and gave phenolic monomers in near theoretical maximum yield (46.4 wt%) with TON up to 431 $mol_{phenols} mol_{Ru}^{-1}$. High selectivity for propyl-substituted guaiacol (Pr-G) and syringol (Pr-S) (84.7 mol%) enabled them to be purified readily. This catalyst demonstrated good stability in recycling experiments as well as after a hydrothermal treatment. The RCF over RuN/ZnO/C left a carbohydrate pulp with high retention because RuN/ZnO/C is noneffective to (hemi)celluloses, which was verified by independent experiments. The reactivity screening of a series of dimeric β-*O*−4 models and a deuterium-incorporated β-*O*−4 polymer offered inspiring insights into the mechanism of current lignin hydrogenolysis.

## Results

### Fabrication and characterization of the RuN/ZnO/C catalyst

The fabrication of the RuN/ZnO/C catalyst started from the deposition of $RuCl_3·3H_2O$ and $Zn(OAc)_2$ with chitosan in a HOAc solution. An aerogel was obtained after freeze drying, which then underwent calcination at 750 °C under $N_2$, affording the RuN/ZnO/C catalyst (Fig. 2a). The contents of Ru and Zn were determined to be 0.12 wt% and 6.2 wt% by inductively coupled plasma atomic emission spectrometry (ICP−AES) (Supplementary Table 1), respectively. Brunauer−Emmett−Teller (BET) adsorption-desorption isotherms revealed that the RuN/ZnO/C catalyst has mesoporous structures with a large pore volume (0.5 $cm^3 g^{-1}$) and surface area (688.6 $m^2 g^{-1}$) (Supplementary Table 1, Supplementary Fig. 1). The $D_1/G$ intensity ratio ($I_{D1}/I_G$) was determined to be 1.7 by Raman spectroscopy, suggesting the presence of defects and disordered structures in the carbon support (Supplementary Fig. 2)[43]. The X-ray diffraction (XRD) patterns exhibited typical peaks ascribed to ZnO (JCPDS No. 36−1451)[42], while there were no detectable Ru-related peaks, probably owing to the extremely low loading and high dispersion of Ru (Supplementary Fig. 3). X-ray photoelectron spectroscopy (XPS) analyses were performed to examine the electron states of Ru, Zn, N,

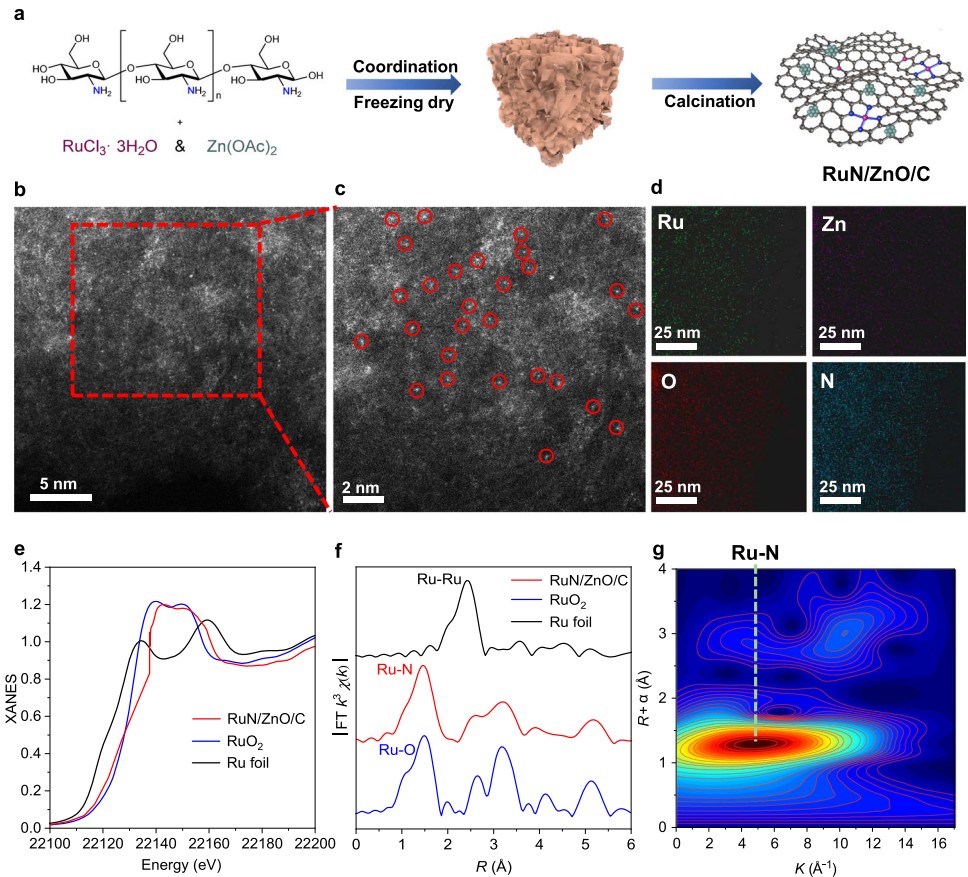

**Fig. 2 | Fabrication and characterization of RuN/ZnO/C. a** Illustration of the fabrication process for RuN/ZnO/C catalyst. **b**, **c** HAADF-STEM images, **d** STEM-EDS elemental mapping analysis of RuN/ZnO/C. **e** Ru K-edge XANES, **f** FT-EXAFS, and **g** WT-EXAFS of RuN/ZnO/C.

O, and C elements (Supplementary Fig. 4). No characteristic signals were detected in the Ru spectrum; two signals resonating at 1044.8 eV and 1021.7 eV were assigned to ZnO in the Zn spectrum[42], being in line with the observations in the XRD patterns. The $N$ $1s$ XPS profile was deconvoluted into four peaks, corresponding to pyridine, pyrrole, graphitic $N$, and oxidized $N$[22,44]. The coordination of Ru atoms with the pyridinic and pyrrolic $N$ species can serve as a stabilization mechanism against leaching during catalyst fabrication and catalytic reactions[46,47]. The morphology of RuN/ZnO/C was characterized by high-angle annular dark-field scanning transmission electron microscopy (HAADF-STEM). As shown in Fig. 2b, c, a large number of ultrasmall brighter spots (0.1–0.2 nm) were assigned to single Ru atoms, with no observation of Ru nanoparticles or small clusters. This is in good agreement with the absence of Ru diffraction peaks in the XRD patterns and XPS spectra. Energy-dispersive X-ray spectroscopy (EDS) mapping showed the uniform distribution of Ru, Zn, N, and O in the RuN/ZnO/C catalyst (Fig. 2d).

To further investigate the structure of RuN/ZnO/C, X-ray absorption fine structure (XAFS) measurements were performed. The Ru K-edge X-ray absorption near edge structure (XANES) spectrum of RuN/ZnO/C is presented in Fig. 2e, together with the spectra of Ru foil and $RuO_2$ for comparison. The energy absorption threshold value of RuN/ZnO/C lied between those of Ru foil and $RuO_2$, indicating that the Ru species carries positive charges. In the Fourier transformed extended X-ray absorption fine structure (FT-EXAFS) spectra, RuN/ZnO/C exhibited an obvious peak at 1.47 Å, which can be attributed to a Ru−N single scattering path (Fig. 2f). The peak corresponding to the Ru−Ru bond at 2.3 Å was not observed in RuN/ZnO/C. R-space fitting of the EXAFS data implied that the first coordination shell of Ru species in RuN/ZnO/C can be well fitted using Ru−N/C single scattering, and no

Ru−Ru single scattering was observed (Supplementary Fig. 6). The distance and coordination numbers of Ru−N/C bonds were refined at 1.95 Å and 4.2 ± 0.7, respectively (Supplementary Table 2). The wavelet transforms (WT) of the $k^3$-weighted EXAFS spectrum displayed only one intensity maximum at 4.7 Å$^{-1}$, which was associated with the Ru−N interaction path (Fig. 2g, Supplementary Fig. 7). According to EXAFS analyses, it could be concluded that Ru elements in RuN/ZnO/C were atomically dispersed on $N$-doped carbon.

### Lignin-derived monomers from RCF of birch

Birch tree, a representative fast-growing hardwood species containing *ca.* 42.3 wt% of cellulose, 19.6 wt% of hemicellulose, and 25.3 wt% of lignin in the stem (Supplementary Table 3), was initially used for RCF. The reaction was performed at 240 °C and $H_2$ (3 MPa at 25 °C) in MeOH for 4 h, where 10 wt% of RuN/ZnO/C catalyst was used (where Ru loading is 0.012 wt% to birch). The RCF of birch chips (2-5 mm) afforded a soluble fraction mainly containing lignin-derived monomers, dimers, and oligomers, as well as, an insoluble fraction composed of cellulose (C6 sugar), hemicellulose (C5 sugar), and catalyst. Gel permeation chromatography (GPC) analyses of the oily product from the soluble fraction gave an average molecular weight ($M_w$) of 325 g mol$^{-1}$, in which two major signals *ca.* 180 and 430 g mol$^{-1}$ were ascribed to phenolic monomers and dimers, and a small broad peak corresponding to oligomers was also observed (Fig. 3e, Supplementary Fig. 9). To identify and quantify products, the soluble fraction was analyzed by gas chromatography (GC) by comparison with corresponding authentic samples. Overall, a 46.4 wt% yield of monophenolics was normalized based on the Klason lignin content in birch (Fig. 3a). This yield achieved an almost theoretical maximum yield of aromatic monomers (*ca.* 49%) by assuming the selective cleavage of β-$O$−4

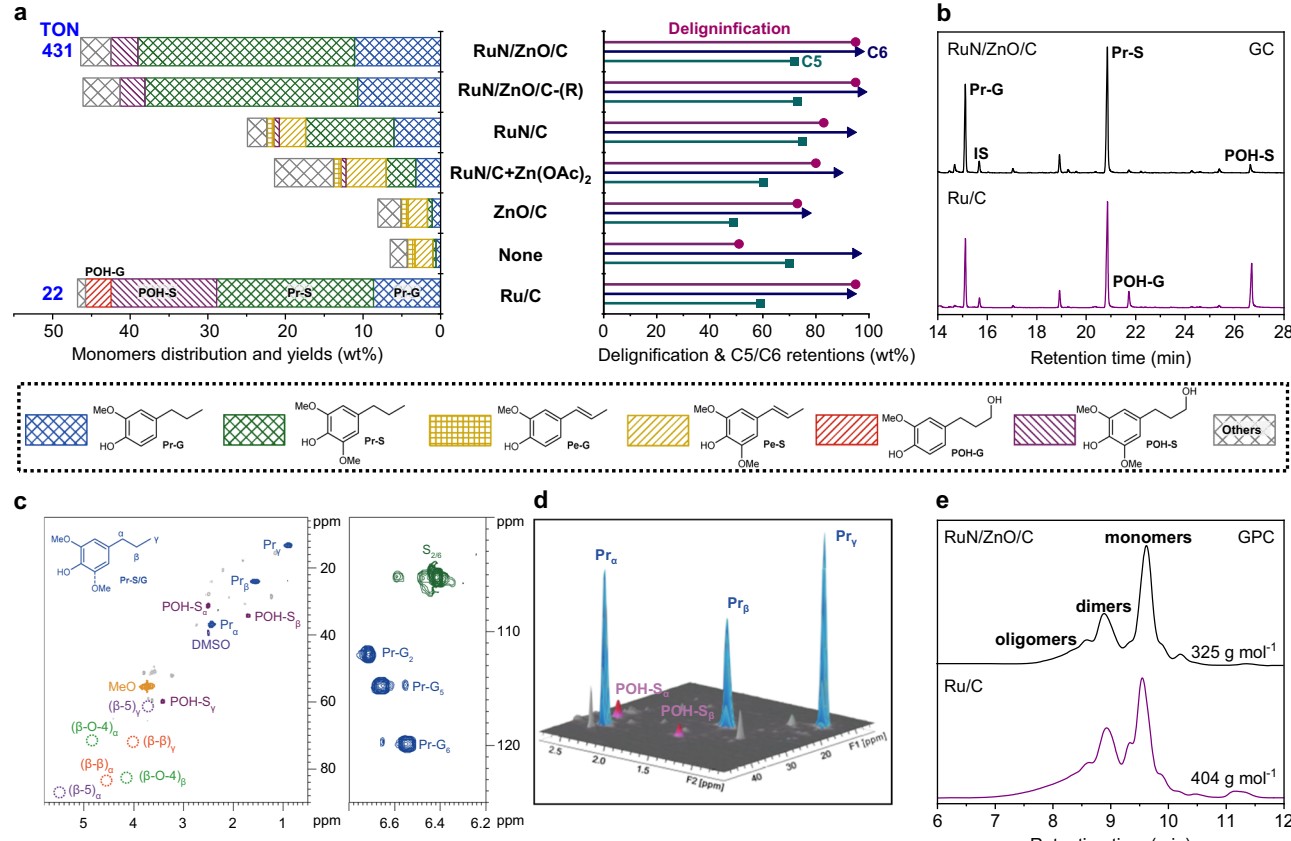

**Fig. 3 | RCF of birch over different catalysts. a** Comparisons of phenolic monomer yields, delignification degrees and C5/C6 retentions over different catalysts. **b** GC spectra of the lignin-derived monomers. **c** 2D HSQC NMR spectra of lignin-derived products from RCF of birch over RuN/ZnO/C (DMSO-$d_6$). **d** Three-dimensional version of the HSQC end-chain region. **e** GPC spectra of lignin-derived products.

Reaction conditions: birch wood (250 mg), catalyst (25 mg, 10 wt%), MeOH (15 mL), 240 °C, $H_2$ (3 MPa at 25 °C, 12 MPa at 240 °C), and 4 h. TON denotes turnover numbers, calculated based on the total number of moles of Ru in the catalyst ($mol_{phenols}\ mol_{Ru}^{-1}$).

linkages lignin component in birch[15]. The TON was calculated as 431 $mol_{phenols}\ mol_{Ru}^{-1}$ based on the total number of moles of Ru in RuN/ZnO/C, which was significantly larger than those from other supported metal catalysts[16,19,22,24,32] (Supplementary Table 13, Supplementary Fig. 33). The ratio of syringol and guaiacyl-derived monomers (S/G) was determined to be 2.4 (mol/mol), similar to the S/G monomer composition in native lignin[8] (Supplementary Table 4). Among the monomers, propyl-substituted syringol (Pr-S, 27.9 wt%) and guaiacol (Pr-G, 11.1 wt%) were identified as two major products, corresponding to 84.7 mol% selectivity of total monomers (Fig. 3b). Interestingly, a small quantity of 4-*n*-propanol syringol (POH-S, 3.5 wt%) was detected, while 4-*n*-propanol-substituted guaiacol (POH-G) was absent. The high selectivity of Pr-S and Pr-G allowed them to be readily purified from the reaction mixture through a chromatographic column (89 mg per gram of birch biomass) (Supplementary Fig. 10), thus paving the way for further transformations[48]. To analyze the dimers, the lignin oils were silylated and characterized on GC-MS. A total of 11 different dimers were identified, among which most had C−C linkages, except for a small number of 4-O-5 dimer (Supplementary Fig. 11)[16,28,29]. These results implied a successful cleavage of most C−O linkages and the reservation of C−C bonds under such a catalyst.

The as-obtained oily product was characterized by 2D HSQC NMR spectroscopy (Fig. 3c). No detectable signals for β-*O*−4 structures remained, indicating an efficient depolymerization of lignin over the RuN/ZnO/C catalyst. A family of cross peaks located at $\delta_C/\delta_H = 36.7/2.42$, 24.1/1.54, and 13.6/0.86 ppm emerged, relating to the propyl end-chain in Pr-S and Pr-G[16,27]. The cross peaks at $\delta_C/\delta_H = 31.4/2.49$, 34.1/1.68, and 60.3/3.41 ppm ascribed to the propanol group in POH-S[17,27]

could also be observed in an inferior version (Fig. 3d). The above results for 2D HSQC NMR spectra were consistent with the observation in GC spectra.

For comparison, several relevant catalysts were screened in the RCF of birch (Fig. 3a). Ru/C is a frequently used catalyst for the RCF of biomass, where the Ru species are generally dispersed on carbon as nanoparticles[16,18,49]. When Ru/C was used at 5 wt% to birch (where Ru loading is 0.25 wt% to birch), a high yield of phenolic monomers was generated (46.7 wt%). The selectivity toward Pr-S and Pr-G (63.8 mol%) was lower than that from RuN/ZnO/C (84.7 mol%). Decreasing the Ru/C dose to 2 wt% led to declines in the yield of monophenols (37.1 wt%) and the Pr-S/Pr-G selectivity (53.3 mol%). The TONs from Ru/C were calculated as 22 and 45 $mol_{phenols}\ mol_{Ru}^{-1}$ with different catalyst dosages, which were far lower than that from RuN/ZnO/C (431 $mol_{phenols}\ mol_{Ru}^{-1}$). The use of ZnO/C resulted in a poor yield of phenolic monomers (8.1 wt%), similar to that from the catalyst-free experiment (6.5 wt%), implying a core role of Ru species in such a transformation, even with extremely low loading. In the case of the RuN/C catalyst without Zn species, drops in phenolic monomers yield (24.9 wt%) and selectivity (69.9 mol%) were both observed, confirming that the introduction of Lewis acid centers, such as Zn species, is beneficial to the hydrogenolysis of β-*O*−4[45]. The combination of RuN/C and Zn(OAc)$_2$ did not promote catalytic performance in birch RCF, suggesting that the anchored ZnO species serve an auxiliary role in lignin hydrogenolysis. A plausible explanation for the synergistic effect between Ru and Zn in RuN/ZnO/C is that the oxyphilic Zn species can tightly bind the OH groups in lignin fragments, thus facilitating the subsequent cleavage of β-*O*−4 structures over Ru species[45].

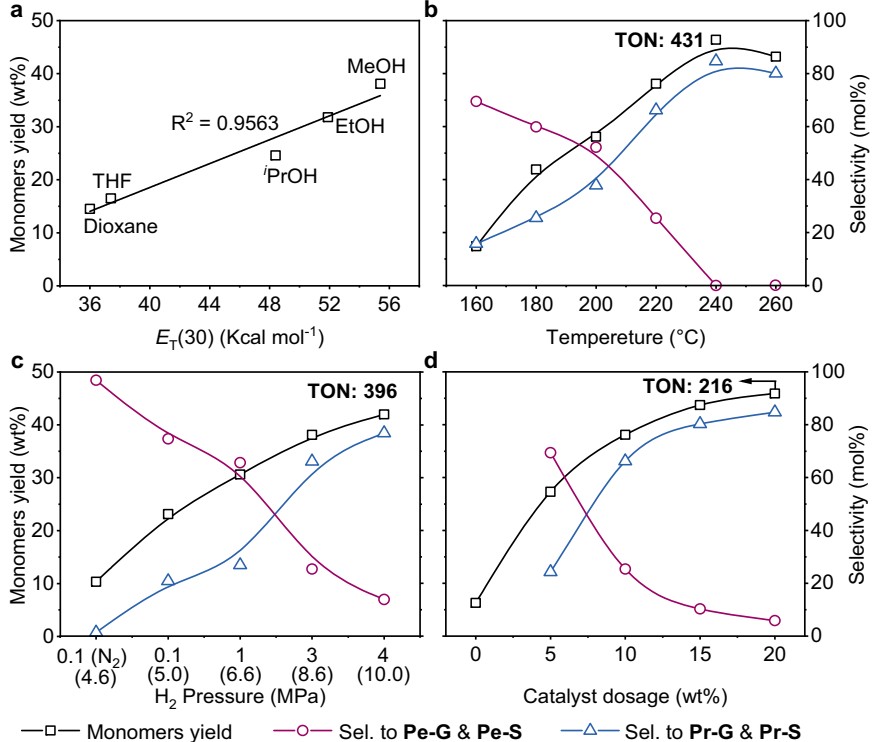

**Fig. 4 | Influences of reaction parameters. a** Solvent effect, conditions: birch wood (250 mg), RuN/ZnO/C (10 wt%), solvent (15 mL), 220 °C, $H_2$ (3 MPa at 25 °C, 8.6 MPa at 220 °C), and 4 h. **b** Reaction temperature effect, birch wood (250 mg), RuN/ZnO/C (25 mg, 10 wt%), MeOH (15 mL), 160–260 °C, $H_2$ (3 MPa at 25 °C), and 4 h. **c** $H_2$ pressure effect, birch wood (250 mg), RuN/ZnO/C (25 mg, 10 wt%), MeOH (15 mL), 220 °C, $N_2$ (0.1 MPa at 25 °C) or $H_2$ (0.1 to 4 MPa at 25 °C, the data in parentheses refer to $H_2$ pressure at 220 °C), and 4 h. **d** Catalyst dosage effect, birch wood (250 mg), RuN/ZnO/C (0–20 wt%), MeOH (15 mL), 220 °C, $H_2$ (3 MPa at 25 °C), and 4 h.

To study the oxidation state of Ru in the reactions, a pre-reduced catalyst RuN/ZnO/C-(R) was prepared by the treatment of $H_2$ in MeOH at 240 °C. In the RCF of birch, RuN/ZnO/C-(R) can produce phenolic monomers in 46.1 wt% yield with 83% selectivity for Pr-G and Pr-S, which is in line with RuN/ZnO/C. We performed XAFS measurements for RuN/ZnO/C-(R). The Ru K-edge XANES spectra showed that the near edge of RuN/ZnO/C-(R) was located between Ru foil and $RuO_2$, revealing the positively charged state of Ru. The EXAFS spectrum of the Ru K-edge in R space displayed a primary peak situated at 2.3 Å, corresponding to Ru–Ru coordination (Supplementary Fig. 8). These results were in accord with the scenarios in HAADF-STEM images, where both atomically dispersed Ru single atoms and Ru nanoclusters were detected (Supplementary Fig. 5). Obviously, most of the single Ru atoms were reduced into nanoclusters during pre-reduction process. The as-formed Ru nanoclusters, which are beneficial to the dissociation of $H_2$ via a homolytic path[50], probably acts as the catalytically active species for lignin hydrogenolysis.

**Analysis and enzymolysis of carbohydrate residue**
In this RCF process, the solid leftovers contained carbohydrate and catalyst. Because the carbohydrate remained in the original framework of biomass (2-5 mm) without evident collapse after RCF, the separation between the catalyst (92% recovery of RuN/ZnO/C) and carbohydrate could be readily achieved through sieving (Supplementary Fig. 25). XRD patterns indicated an increase in the relative crystallinity index from 41.8% of birch to 50.9% of carbohydrate pulp (Supplementary Fig. 26). Biomass compositional analysis suggested that 95% of lignin was removed from the carbohydrate residue, and cellulose (C6) and hemicellulose (C5) were reserved as solid phases in 96% and 72% by weight, respectively (Fig. 3a). The established carbohydrate retention over RuN/ZnO/C was superior to those from Ru/C (C6, 93%; C5, 59%)

and comparable to those from non-catalytic process (C6, 95%; C5, 70%). To evaluate the reactivity of carbohydrate with RuN/ZnO/C, independent experiments employing commercial cellulose and hemicellulose biopolymers were treated with or without RuN/ZnO/C (Supplementary Fig. 28). For cellulose reactions, 82 wt% of cellulose solid was recovered after catalytic treatment, while non-catalytic process gave carbonized cellulose in 72 wt% retention. The treatment of hemicellulose resulted in similar solid retention rates of 37 wt% and 34 wt% with and without catalyst, respectively. These results hinted that current RuN/ZnO/C is a highly specific catalyst for lignin transformation rather than carbohydrates during the RCF process.

Upon cellulase treatment of the as-obtained carbohydrate residue, a high yield of glucose (88%) from cellulose was achieved, which is far higher than the yield obtained with the use of ball-milled birch biomass (35.6%) (Supplementary Fig. 29). This is because the complete removal of lignin by the RCF process provided larger surface areas in carbohydrate residues for the access of enzymes but crucially broke the biomass recalcitrance[18,30]. Hence, RCF over RuN/ZnO/C constituted an economical pathway for the valorization of lignin as well as the fractionation of lignocellulosic biomass. The enzymolysis left a hemicellulose solid, corresponding to 66 wt% of the initial hemicellulose in birch biomass. Overall, RCF of birch and subsequent enzymatic transformation generated phenolic monomers, glucose, and hemicellulose, corresponding to 46.4 wt% of lignin, 84 wt% of cellulose, and 66 wt% of hemicellulose, respectively. Thereby, 69 wt% of detectable organic carbon in birch biomass has been consequently converted into valorized products (Supplementary Fig. 30).

**Parameters effects**
A range of solvents were screened in the RCF of birch in the presence of RuN/ZnO/C at 3 MPa $H_2$ and 220 °C (Fig. 4a, Supplementary Table 5,

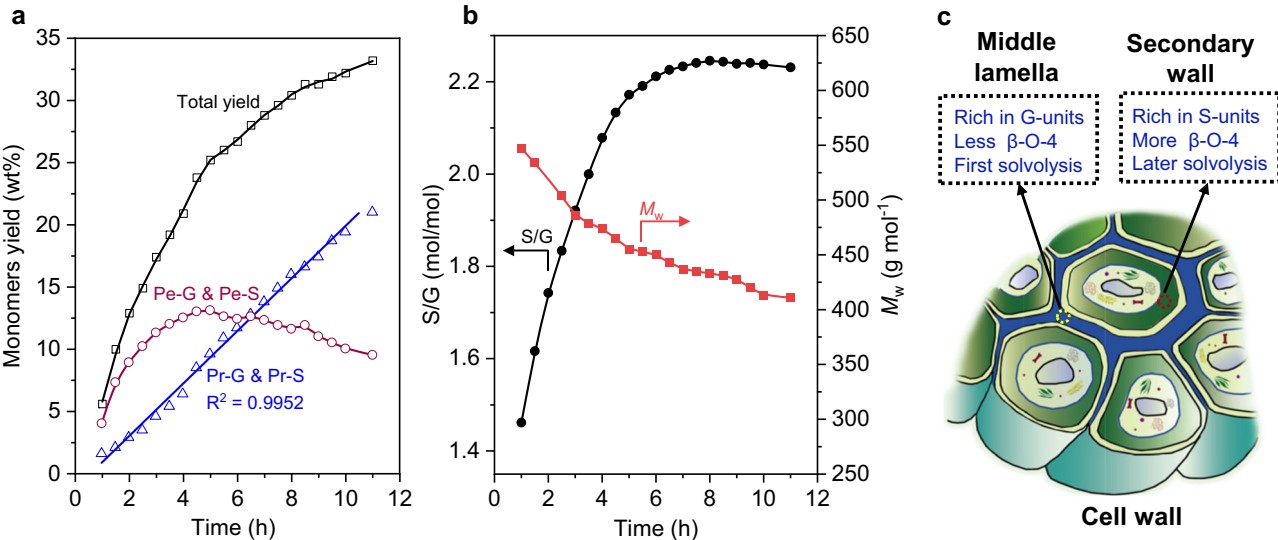

**Fig. 5 | Time profile of the birch RCF reaction over the RuN/ZnO/C catalyst.**
**a** Cumulative monomer yields. **b** S/G ratios of monomers and average molecular
weights of lignin oils. **c** Illustration of the cell wall. The reaction was performed in a
300 mL autoclave containing birch sawdust (3 g), RuN/ZnO/C (300 mg), MeOH
(150 mL), 7.2 MPa $H_2$ at 200 °C.

Supplementary Fig. 13). Compared to MeOH, the use of other solvents,
such as EtOH, $^i$PrOH, THF, and dioxane, led to decreases in monomeric
phenols, the selectivity to Pr-S and Pr-G, and the degree of delignifi-
cation. Propenyl-substituted syringol (Pe-S) and guaiacol (Pe-G)
emerged (Supplementary Table 5) and may serve as intermediates to
Pr-S and Pr-G[20]. The total monomers yields, as well as the degrees of
delignification, approximately followed linear relationships with the
solvent polarity ($E_T(30)$)[51]. More polar solvents, such as MeOH, more
easily penetrate the lignocellulosic matrix and extract lignin, thus
promoting non-catalytic delignification[51]. During catalytic cleavage of
the β-$O$-4 step, MeOH may serve as a hydrogen donor to assist the
reduction process[49]. In this context, the procedures involved in RCF of
birch, that is, the disassembly of lignin and subsequent fragmentation
of lignin, both benefit from higher solvent polarity.

We then investigated the effects of the reaction temperature,
catalyst dosage, and $H_2$ pressure on the RuN/ZnO/C catalyst in MeOH
(Fig. 4b–d and Supplementary Tables 6–9). At a low reaction tem-
perature, decreased monophenols yields and selectivity to Pr-S and Pr-
G were both observed. For example, RCF at 160 °C generated only
7.4 wt% yield of monomers with 69.5% selectivity to unsaturated Pe-S
and Pe-G. The highest yield of phenolic monomers was obtained at
240 °C, with a completely saturated end-chain. The reaction per-
formed at 260 °C led to a slight decrease in monomers yield, probably
due to recondensation of the resulting monomers at high tempera-
ture. The variations in $H_2$ pressure and catalyst dosage both presented
positive correlations with the yields of monophenolics and the selec-
tivity to Pr-S and Pr-G. A linearity between monophenols yields and the
mole fraction solubility of $H_2$ in MeOH was observed (Supplementary
Fig. 17), indicating that sufficient $H_2$ transferred from the gas to the
liquid phase is necessary for current lignin hydrogenolysis. Of note, a
nonnegligible amount of propanol syringol (POH-S, 0.2-3.7 wt% yield)
was detected in all cases, whereas propanol guaiacol (POH-G) was
not found.

## RCF time course

To monitor the RCF of birch as a function of reaction time, a large-scale
reaction (3 g of birch and 300 mg of RuN/ZnO/C) was performed in
MeOH at 200 °C with 7.2 MPa $H_2$ partial pressure (Fig. 5, Supplemen-
tary Table 10). An aliquot was picked up every half hour for analyses
during the experiment. Phenolic monomers were formed steadily until

up to 33.2 wt% yields over the course of the 11 h reaction. The forma-
tion of Pr-S and Pr-G was correlated with reaction time in a nearly linear
fashion. Furthermore, the accumulation of propenyl-chained products
(Pe-S and Pe-G) reached a maximum in the first 5 h and then gradually
decreased (Fig. 5a). These results again confirmed that Pr-S and Pr-G
were derived from Pe-S and Pe-G through hydrogenation of the pro-
penyl end-chain.

The S/G ratios of as-resulted monomers over the reaction times
showed a similar trend to that at total monomers yields (Fig. 5b). The
scenario of smoothly increasing the S/G ratios to stability was in line
with a previous report, wherein RCF of poplars were performed over a
Ni/C catalyst in a flow-through reactor[29]. The relatively low abundance
of syringol derivatives at the incipient stage should be associated with
lignin diffusion in the plant cell wall; that is, lignin in the middle lamella
having fewer methoxyl groups (rich in G-units) than lignin in the sec-
ondary wall (rich in S-units)[52,53] could be preferentially and rapidly
solvolyzed into MeOH for depolymerization (Fig. 5c). This deduction
was also validated by the concentrations of Pe-G and Pe-S, which
peaked at 3 h and 5 h, respectively (Supplementary Fig. 19). The aver-
age molecular weights ($M_w$) of the as-picked lignin oily samples pre-
sented a descending trend, as seen in 1 h and 11 h, and the $M_w$ values
were measured as 547 g mol⁻¹ and 411 g mol⁻¹, respectively (Supple-
mentary Fig. 20). A possible explanation is that the earlier released
middle lamella lignin having fewer β-$O$-4 units (than the later released
secondary wall lignin)[54] is hard to be depolymerized completely, thus
leading to larger $M_w$ of lignin oils in the early stage of RCF. The current
$M_w$ tendency differed from the observation of RCF of poplars over Ni/
C, where the $M_w$ first increased (up to 2000 g mol⁻¹) and then
decreased over the reaction time[29]. Thus far, it was implied that the
dissolved lignin fragments from the biomass matrix could be depoly-
merized without any hesitation over RuN/ZnO/C, thus avoiding the
recondensation reactions.

## Stability and reusability of RuN/ZnO/C

For industrial applications, a heterogeneous catalyst should have
strong stability to guarantee reusability, minimize the recovery of
metal active species and avoid contamination for economic reasons.
To assess the survivability of RuN/ZnO/C under harsh conditions, it
was treated by a hydrothermal (H.T.) process at 200 °C for 72 h. The
hydrothermally treated RuN/ZnO/C still exhibited constant catalytic

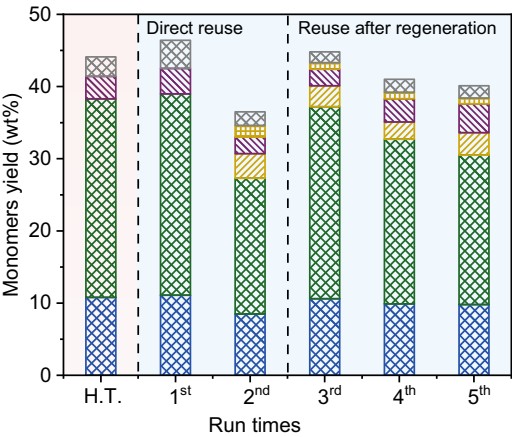

**Fig. 6 | Stability and reusability of RuN/ZnO/C catalyst.** Reaction conditions: birch wood (250 mg), RuN/ZnO/C (25 mg, 10 wt%), MeOH (15 mL), 240 °C, H₂ (3 MPa at 25 °C), and 4 h.

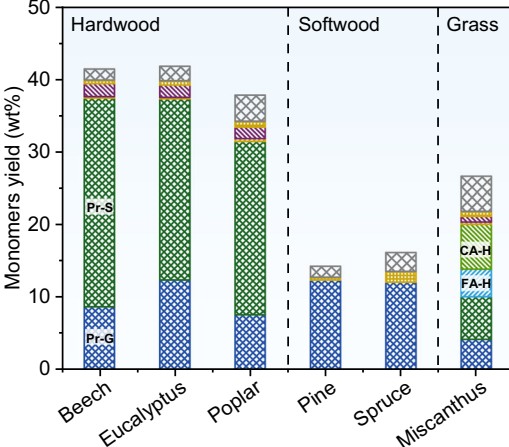

**Fig. 7 | RCF of various biomass sources.** Reaction conditions: biomass sawdust (250 mg), RuN/ZnO/C (25 mg, 10 wt%), MeOH (15 mL), 240 °C, H₂ (3 MPa at 25 °C, 12 MPa at 240 °C), and 4 h. CA-H and FA-H represent methyl 3-(4-hydroxyphenyl) propanoate and methyl 3-(4-hydroxy-3-methoxyphenyl)propanoate, respectively.

performance in the RCF of birch, and gave monophenolic compounds in 44.1 wt% yield with 88.2% selectivity to Pr-S and Pr-G (Fig. 6). Another atomically dispersed Ru catalyst, Ru/ZnO/C-(MOF) without coordinated nitrogen element, which was generated by the in situ anchoring Ru in MOFs and the pyrolysis strategy[42], was tested by hydrothermal treatment. Ru/ZnO/C-(MOF) gave 35 wt% monomeric phenolics in the RCF of birch, whereas H.T.-treated Ru/ZnO/C-(MOF) gave a decreased yield (30 wt%) (Supplementary Table 4). In this context, the coordinated N moieties in RuN/ZnO/C play an important role in the stabilization of active Ru species.

To evaluate the reusability of RuN/ZnO/C, the spent catalyst isolated from the carbohydrate pulp by sieving was submitted for birch RCF directly (Fig. 6, Supplementary Fig. 25). In the 2nd run, a decreased catalytic performance was observed, generating monophenols in 36.5 wt% yield with 75.4% selectivity to Pr-S and Pr-G. ICP–AES analyses gave a 0.12 wt% content of Ru, being in line with the fresh sample (Supplementary Table 1); this, together with no detectable Ru element in the soluble fraction, implied that there was no Ru leaching during the RCF process. The slightly decreased content of Zn in the spent RuN/ZnO/C (4.9 wt%) and the detectable Zn in the liquid phase indicated leaching of Zn. Compared to the fresh catalyst, the specific surface area and pore volume of the spent RuN/ZnO/C were decreased. In this context, the spent catalyst was regenerated through calcination at 500 °C, by which the porous structures were resumed (Supplementary Table 1, Supplementary Fig. 1). We then used the regenerated RuN/ZnO/C for the 3rd run, which gave 44.8 wt% yields of monomers and 83.7% selectivity to Pr-S and Pr-G, suggesting almost complete recovery of catalytic performance. These results implied that the catalyst deactivation may be mainly because of the fouling of metal/ support and that a small loss of Zn does not influence the catalytic performance. Further experiments confirmed that there was no significant loss of catalytic performance by performing the regenerated catalyst after 5 runs (Fig. 6).

### RCF of various biomass
We further evaluated the RCF performance of other fast-growing wood and grass biomass over RuN/ZnO/C (Fig. 7, Supplementary Table 12). The hardwoods, such as beech, eucalyptus, and poplar, gave phenolic monomers in near theoretical maximum yields (37.9–41.9 wt%) with Pr-S and Pr-G as the dominant products (85.3–91.6% selectivity). In the case of poplar biomass, unique 4-methylhydroxybenzoate (0.9 wt%) was also detected[55]. In contrast to hardwood, lignin in softwoods (such as pine and spruce) is mainly composed of G-units with less cleavable β-O-4 linkages[8,16], which upon treatment with RuN/ZnO/C, afforded

monomeric phenols in 14.3–16.2 wt% yields. As expected, a simplex product Pr-G was detected with 86.7% and 74.9% selectivity, which means that it could be purified readily. When miscanthus, a perennial grass, was treated with RuN/ZnO/C, phenolic monomers were generated with a 26.7 wt% combined yield. In addition to Pr-G (4.2 wt%) and Pr-S (5.9 wt%), two specific phenolic monomers (CA-H, 6.2 wt%; FA-H, 3.8 wt%) from the p-coumaric and ferulic acid units were also generated. 2D HSQC NMR spectroscopies of lignin oils from pine, spruce and miscanthus plants indicated that there were no existing β-O-4 structures, illustrating the fullest cleavage of β-O-4 units (Supplementary Fig. 24).

### Mechanistic insights
To investigate the catalytic hydrogenolysis mechanism of lignin by the RuN/ZnO/C catalyst, the reactivity of a series of lignin model compounds was screened with RuN/ZnO/C, and the results are presented in Fig. 8 (see also Supplementary Fig. 31). During lignin depolymerization, phenolic groups would be exposed after the cleavage of internal β-O−4 units. Hence, model compounds **1** and **2** were used to mimic phenolic units of lignin. Catalytic hydrogenolysis of **1** gave Pr-G (73%) as a major end-chained product, along with the observation of Pe-G (9%) (reaction a). This product distribution was similar to those from the RCF of wood biomass. The high conversion of Pe-G to Pr-G (reaction b) and the nonactivity of POH-G (reaction d) confirmed again that Pe-G acts as an intermediate toward Pr-G. The reaction of coniferyl alcohol with RuN/ZnO/C afforded Pr-G (57%) and Pe-G (12%), and a non-negligible amount of POH-G (15%) (reaction c), differing from the scenarios observed in **1** and lignin biopolymer reactions, where no POH-G was detected. Therefore, it was deduced that the $C_γ$–OH bond may be cleaved at the β-O−4 dimeric stage[27]. The hydrogenolysis of **2** with a syringyl unit formed Pr-S in 87% yield (reaction e). Interestingly, POH-S (3%) was also detected here, similar to the scenario with RCF of birch. This may be because S-units bearing two methoxy groups tend to depolymerize[9,16], thus releasing a small amount of sinapyl alcohol before the entire removal of γ-OH in β-O−4 structures. In the case of the nonphenolic β-O-4 dimer **3** that can model the internal units of lignin, the RuN/ZnO/C-catalyzed depolymerization reaction offered propyl- and propenyl-substituted products in 36% and 16% yields, respectively (reaction f). The relatively low monomers yield from **3** should result from its relatively high energy barrier of β-O-4 destruction[56].

Dimeric β-O-4 models lacking the γ-CH₂OH unit, such as compounds **6** and **8**, can efficiently give depolymerized products in high

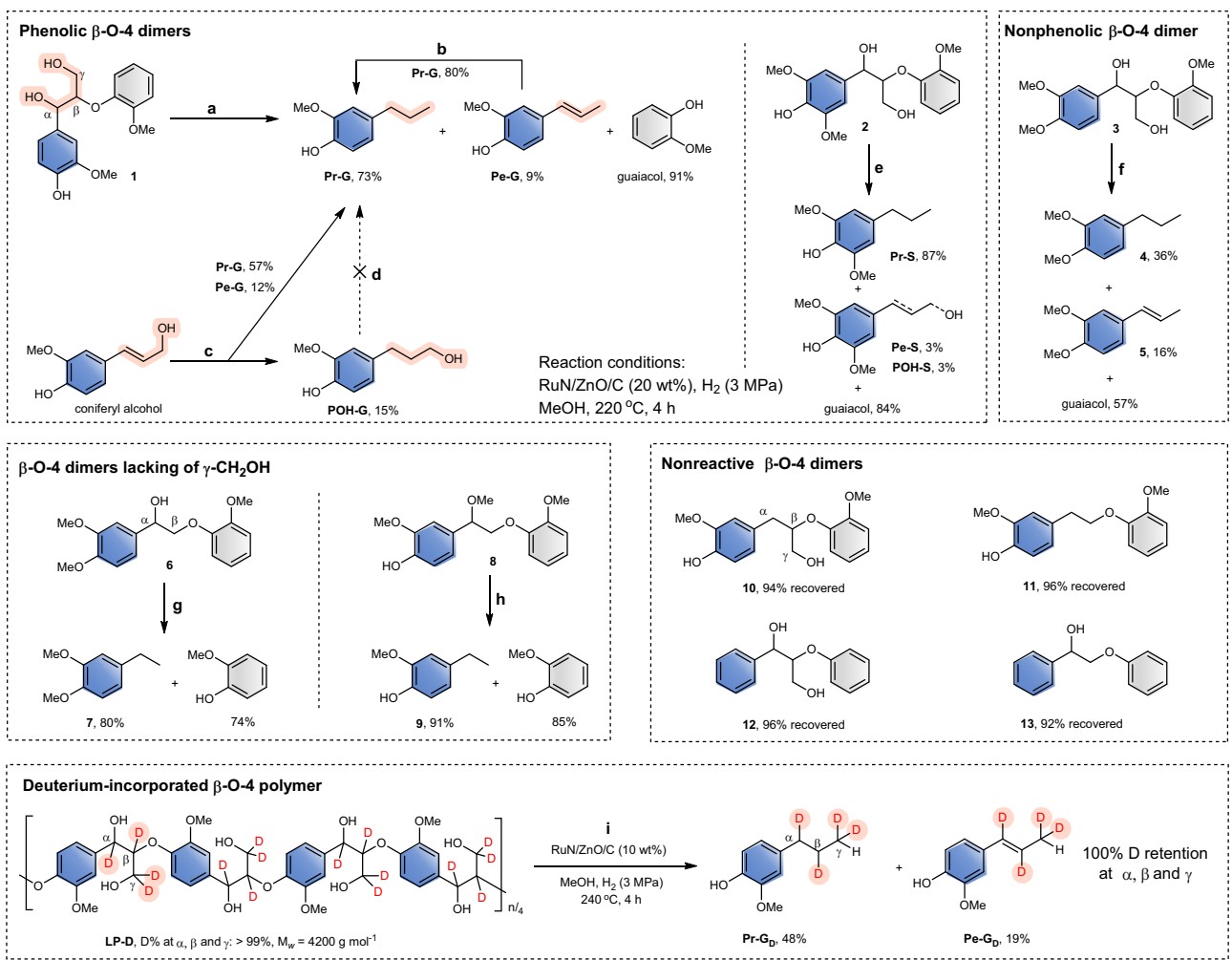

**Fig. 8 | Reactivity of β-*O*-4 lignin mimics with RuN/ZnO/C.** Reaction conditions for the other dimers: substrate (25 mg), RuN/ZnO/C (5 mg, 20 wt%), MeOH (15 mL), 220 °C, H$_2$ (3 MPa at 25 °C), and 4 h.

yields, indicating that γ-OH does not influence the cleavage of β-*O*-4 units (reactions g and h). In contrast, no depolymerized reactions occurred in the case of β-*O*-4 models without α-OH, such as model compounds **10** and **11**. This scenario accorded with the previous conclusion drawn from Pd/C-catalyzed lignin hydrogenolysis; that is, compound **10** (or analogs) is recognized as a side-product rather than an intermediate in catalytic hydrogenolysis of **1**, and the C$_β$–O bond in **10** cannot be cleaved during hydrogenolysis[45,57]. To our surprise, compound **12**, having an integral β-*O*-4 skeleton but lacking aromatic OH and MeO groups, could not undergo the cleavage reaction, where 96% of **12** was recovered under the RuN/ZnO/C catalyst. A similar scenario for **13** was also observed. These results implied that oxygenated moieties at the aromatic ring not only act as a key issue for lignin biosynthesis via oxidative polymerization[7,8] but also play a crucial role in the reductive depolymerization of lignin with RuN/ZnO/C.

To gain more insight into the mechanism of this transformation, a synthetic β-*O*-4 polymer with deuteriums anchored at the α, β and γ positions (LP-D)[45], was treated with RuN/ZnO/C (Fig. 8). This transformation generated deuterated guaiacol derivatives Pr-G$_D$ and Pe-G$_D$ in 48 wt% and 19 wt% yields, respectively. The Pr-G$_D$ could be purified from the reaction mixture. Characterization of Pr-G$_D$ by $^1$H NMR spectroscopy showed that the deuteriums at the α, β and γ positions remained intact (Supplementary Fig. 32). Mass analyses also confirmed the entire reservation of deuteriums in Pr-G$_D$ and Pe-G$_D$ by comparison

with authentic samples. These results indicated that the protons in β-*O*-4 linkages exist outside of the cleavage of C$_α$–OH and C$_β$–O bonds steps, thus categorically excluding the feasibility of pathways involving dehydrogenation and/or dehydration reactions under such conditions. Taking into consideration Pe-G with a C$_α$=C$_β$ bond serving as intermediates, together with no participation of α-H and β-H during β-*O*-4 cleavage, we deduced that the cleavages of C$_α$–OH and C$_β$–O bonds in β-*O*-4 moieties should occur synchronously, probably via a concerted hydrogenolysis/elimination mechanism[45,49].

## Discussion

The results described herein demonstrate the RCF of lignocellulosic biomass by an extremely low-loaded, highly dispersed Ru catalyst supported on chitosan-derived *N*-doped carbon. This catalyst afforded theoretical maximum yields of phenolic monomers from lignin, along with high preservation of cellulose and hemicellulose amenable to enzymatic valorization. The current RuN/ZnO/C catalyst outperformed the commercial Ru/C catalyst regarding the atomic economy of Ru (twenty-fold TON) and selectivity, benefiting from the atomic size of Ru and the promotion effect of ZnO. This catalyst also demonstrates tough stability in recycling RCF reactions or after a hydrothermal process, correlating with the coordination of Ru and N species. Characterizations and catalytic experiments proposed that Ru nanoclusters generated from signal atoms under such a condition may

serve as the catalytically active species for lignin hydrogenolysis. Mechanistic studies by lignin model compounds supported the crucial role of α-OH and aromatic O-species in the β-O-4 structure, and that also excluded the feasibility of pathways containing dehydrogenation and/or dehydration reactions. This work represents a significant advancement in developing cost-effective, high-performance, and tough stable catalysts for lignin depolymerization as well as the fractionation and fullest utilization of lignocellulose biomass components.

## Methods
### Chemicals
Chitosan (medium viscosity, 200–400 mPa·s), zinc acetate (99.9%), and acetic acid (99.5%) were purchased from Energy Chemical. Ruthenium chloride trihydrate was purchased from Shanxi Kaida Chemical. Commercial cellulase (NS22086, 1000 BHU (2) g$^{-1}$) was obtained from Novozymes. Birch wood (*Betula*), beech wood (*Zelkova*), eucalyptus grandis (*Eucalyptus*), poplar (Populus tomentosa carr.), pine wood (*Pinus*), spruce wood (*Picea*), and miscanthus (*Miscanthus lutarioriparius*) were screened into powders in size of 2–5 mm, extracted by toluene/ethanol for 5 h and dried at 60 °C before use. Dimeric and polymeric lignin model compounds were prepared following the literature reports[45,49]. All reagents and solvents were used as received without additional purification.

### Preparation of RuN/ZnO/C catalyst
Chitosan (1 g) and HOAc (1 mL) were mixed in deionized water (50 mL) under vigorously stirring to form a homogenous dispersion. The aqueous solutions of RuCl$_3$·3H$_2$O (2 mg in 40 mL H$_2$O) and Zn(OAc)$_2$ (0.8 g in 40 mL H$_2$O) were injected into chitosan solution under stirring at 60 °C, respectively. After the mixture was stirred at 60 °C for 24 h, half of the water was removed by the rotary evaporation. The left water was removed through freezing at −18 °C and followed vacuum treatment, from which an aerogel was obtained as catalyst precursor. Calcination of as-prepared aerogel at 750 °C in the tube furnace for 2 h under N$_2$ flow gave the RuN/ZnO/C catalyst as black solid.

### Hydrogenolysis of biomass sawdust
In the typical RCF experiment, birch chips (250 mg, 2–5 mm), RuN/ZnO/C catalyst (25 mg) and MeOH (15 ml) were charged into the 50 mL stainless steel reactor (Parr Instruments & Co.). The reactor was purged with N$_2$ to remove O$_2$, and then was pressurized with H$_2$ to the desired pressure at room temperature. The reaction mixture was heated to different temperatures and kept for a certain time under magnetic stirring. Afterwards, the reactor was cooled and depressurized at room temperature. The soluble fraction and solid phase (carbohydrate pulp and catalyst) were separated through filtration with a 0.22 μm filter.

### Time profile experiment
The time profile experiment was performed in a parr autoclave (300 mL) equipped with a mechanical stirrer, and a constant pressure complementary H$_2$ device. The reactor was loaded with birch chips (3 g), RuN/ZnO/C catalyst (300 mg), MeOH (150 mL), and 1,3,5-trimethoxybenzene (75 mg, internal standard), which was then purged with N$_2$ and filled with H$_2$. The experiment was performed at 200 °C with stirring at 300 r.p.m., and the H$_2$ pressure was kept at 7.2 MPa. An aliquot (ca. 1 mL) was picked up per half an hour, which was then analyzed on GC and GPC.

## Data availability
The source data underlying Figs. 2e, f, 3a, b, e, 4a–d, 5a, b, 6, and 7 and Supplementary Figs. 1–4, 6, 8, 9, 11, 13–18, 20–23, 26, 29, 31, 32 is provided as a Source Data file. The data that support the findings of this study are available from the corresponding author upon reasonable request.

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

## Acknowledgements

G.S. acknowledges the National Natural Science Foundation of China (31971607) and the National Key Research and Development Program of China (2019YFC1906700). Y.F. acknowledges the National Key Research and Development Program of China (2021YFC2103703). S.W. acknowl-edges the Fundamental Research Funds for the Central Universities (BLX202133). All NMR experiments were carried out at the BioNMR facility, Tsinghua University Branch of China National Center for Protein Sciences (Beijing). We thank Dr. Ning Xu for his assistance in NMR data collection.

## Author contributions

G.S., Z.L., S.W., and Y.F. conceived the project. Z.L. designed and syn-thesized the catalysts, and performed the catalysts characterizations and catalytic experiments. Z.L. and H.L. carried out the catalytic hydrogenolysis of lignin model compounds. Xue.G. assisted in XAFS data analyses. Xua.G. performed the hydrothermally treated catalysts. G.S. and Z.L. wrote the manuscript. All authors discussed the results and commented on the manuscript.

## Competing interests

The authors declare no competing interests.

## Additional information

**Supplementary information** The online version contains

supplementary material available at

Shuizhong Wang, Yunming Fang or Guoyong Song.

