## [Peer Review File · Nature Communications]

Title: Rational highly dispersed ruthenium for reductive catalytic fractionation of lignocelluloseREVIEWER COMMENTS

Reviewer #1 (Remarks to the Author):

The authors report on the use of a single atom/atomically dispersed Ru catalyst for the lignin-first reductive catalytic fractionation of various kinds of woody biomass feedstock, with particular emphasis on birch. Close to theoretical monomer yields are obtained, with very high selectivities to the propylguaiacol and syringol monomers. Overall, the study is very comprehensive and well executed. The catalyst is extensively characterized, including the necessary HAAFD-STEM and XAS analyses, and the lignin-first studies are extensive, detailing full biomass utilization, feedstock variation, catalyst reuse and some model studies.

The catalyst is synthesized from a chitosan C-source with Zn(OAc)₂ added to give the additional Lewis acidity, a feature of the catalyst that later proves quite essential. It should be noted that the authors previously (in Nat Commun, ref 42) reported on a MOF-derived atomically dispersed Ru/ZnO/C catalyst and used this system in C-lignin hydrogenolysis. In the previous contribution, catalyst stability and reuse was not covered. In the present study, the remaining N-functional groups in the carbon support are suggested to provide additional stability to the single Ru sites, which would then be the novel aspect of this particular catalyst system.

Overall, the high, close to theoretical yields have been obtained before in reported RCF efforts with traditional nanoparticulate carbon-supported precious metal catalysts. The impact of the paper then hinges on the combination of efficient precious metal use and the intrinsic stability of the catalyst, i.e. the ability for extensive reuse. The authors report on a regeneration method, which restores activity to close the original values, at least for one run, after this a small gradual drop in activity is seen in Fig 6. Emphasis is put on the stability of the Ru single sites, but the elemental analysis reported in the SI do show considerable leaching of the Zn Lewis acid component, likely simply due to leaching. This is not commented on much in the main text. I am therefore hesitant as to if the main advance, i.e. the development of a stable atomically dispersed catalyst for RCF, is sufficiently demonstrated by the results. This would require a comparison with the MOF derived catalyst, a more extensive study on the fate and influence of the (soluble?) Zn component and more extensive reuse.

As noted, the study is very well done and very extensive and would provide a very interesting addition to the literature on lignin first; impact and novelty wise a stronger case needs to be made in my opinion to be suitable for publication in Nature Commun.

Please find some additional comments below:

- While not part of course of the assessment of the merits of the paper, the manuscript is in need of a very careful and extensive language check
- The authors should more explicitly compare their results with (own) literature; this holds for the catalyst synthesis part, but also for the use of a Lewis acid cocatalyst (cf. the system by Abu-Omar which

contains Pd+Zn)

- It would be interesting to provide more information on the dimer/trimer/oligomer components of the lignin oil obtained; the group of Sels have recently reported extensive analyses that could of use here.

Reviewer #2 (Remarks to the Author):

The authors present a new Ru catalyst for reductive catalytic fractionation of lignocellulosic materials. The results are fascinating and provide further insight into the relationships between catalyst and lignin's active stabilisation. The experimental evidence supports the claims. However, it is unclear how the authors could separate the catalyst from the pulp (on page 15, "To evaluate the reusability of RuN/ZnO/C, the spent catalyst isolated from the carbohydrate pulp was submitted for birch RCF directly"). The experimental details need to be presented in Experimental since catalyst recovery in RCF is still very problematic.

In summary, it is an exciting manuscript with very nice results.

Reviewer #3 (Remarks to the Author):

The authors present a nice study on developing a Ru single atom catalyst for reductive catalytic fractionation of biomass. The manuscript needs to be edited for minor grammatical mistakes (e.g., articles). The manuscript handles the synthesis of a new catalyst, detailed characterization of the catalyst, the activity of the catalyst in the hydrogenolysis of birch sawdust, effect of reaction conditions on hydrogenolysis yield, effect of solvent and comparison of different biomass sources. Authors put effort on completing the picture of several aspects of the new catalyst and compared it to standards catalysts. It is a very dense manuscript, however there are some points need to be clarified. The manuscript could be considered for Nature Communications after addressing the issues listed below:

1. For the determination of the biomass composition, NREL procedure was used. The total composition obtained was found to be 87.2%. This looks a bit low for total composition analysis. Generally, it would be 95+% by the NREL protocol. Why this is so low? Did you grind the biomass to the conditions?

2. Is the catalyst pre-reduced before the reaction? If not, the authors should comment on the catalyst oxidation state under reaction conditions. One would expect methanol reforms and reduces the catalyst; however, catalyst characterization does not clarify this. Authors should comment of this important aspect of the catalyst.

3. Related with the question 2, do you have any EXAFS study related with the reduction of Ru atoms? Did you perform the experiments before and after reduction and see the differences?

4. Regarding Figure 4c: Liquid phase reactions are expected not to be affected by the pressure change. At the pressures studied, majority of the reaction mixture is in the gas phase. The authors should provide thermodynamic calculations for the conditions used. Also, are these the final pressures or the set pressures at the room temperature?

5- On page 22, the procedure for the time profile experiment is described. It is specified that the reaction pressure was kept at 3 MPa at 200 °C. According to the phase diagram, under these conditions, methanol is expected to be completely in gas phase. Is 3 MPa your operating pressure? If so, this complicates the reaction system, and may affect the reliability of the results.

6- Supplementary Information, page 4 describes the analytical procedure for calculating the monomer yield. In the first paragraph, instead of "...available standards or independent synthesis.", "...available standards or independently synthesized standards." would be easier to understand.

7- In the formula given in supplementary information, on page 4, what does M (total monomers) correspond to? Is this the mass of the detected and quantified monomers (e.g. Pr-G, Pe-G, POH-S,...) or is it referring to the mass after converting the moles of quantified monomers to the moles of monolignols (coniferyl alcohol, sinapyl alcohol) and their corresponding mass? The equation should be referring to the mass of the monolignols found in the biomass to determine the weight percent of Klason lignin.

8- Could authors provide the details of GC/MS method (temperature, ramp, calibration range)?

9- On the supplementary information, page 5, what is the calibrated MW range? Was a calibration kit used? If so, what is the part number and supplier?

10- Table 1 of supplementary information lists the ICP-AES results of the fresh and spent catalyst. It looks like some of the Zn leaches. Was that confirmed by analyzing the liquid mixture? The authors should comment on the effect of Zn leaching on catalyst stability.

11- On page 19, line 458, Fig. 7 should be Fig. 8.

12- The references are not in the same format.

The point-by-point response to the reviewers' comments

Reviewer #1 (Remarks to the Author):

The authors report on the use of a single atom/atomically dispersed Ru catalyst for the lignin-first reductive catalytic fractionation of various kinds of woody biomass feedstock, with particular emphasis on birch. Close to theoretical monomer yields are obtained, with very high selectivities to the propylguaiacol and syringol monomers. Overall, the study is very comprehensive and well executed. The catalyst is extensively characterized, including the necessary HAAFD-STEM and XAS analyses, and the lignin-first studies are extensive, detailing full biomass utilization, feedstock variation, catalyst reuse and some model studies.

The catalyst is synthesized from a chitosan C-source with $\text{Zn}(\text{OAc})_2$ added to give the additional Lewis acidity, a feature of the catalyst that later proves quite essential. It should be noted that the authors previously (in Nat Commun, ref 42) reported on a MOF-derived atomically dispersed Ru/ZnO/C catalyst and used this system in C-lignin hydrogenolysis. In the previous contribution, catalyst stability and reuse was not covered. In the present study, the remaining N-functional groups in the carbon support are suggested to provide additional stability to the single Ru sites, which would then be the novel aspect of this particular catalyst system.

Overall, the high, close to theoretical yields have been obtained before in reported RCF efforts with traditional nanoparticulate carbon-supported precious metal catalysts. The impact of the paper then hinges on the combination of efficient precious metal use and the intrinsic stability of the catalyst, i.e. the ability for extensive reuse. The authors report on a regeneration method, which restores activity to close the original values, at least for one run, after this a small gradual drop in activity is seen in Fig 6. Emphasis is put on the stability of the Ru single sites, but the elemental analysis reported in the SI do show considerable leaching of the Zn Lewis acid component, likely simply due to leaching. This is not commented on much in the main text. I am therefore hesitant as to if the main advance, i.e. the development of a stable atomically dispersed catalyst for RCF, is sufficiently demonstrated by the results. This would require a comparison with the MOF derived catalyst, a more extensive study on the fate and influence of the (soluble?) Zn component and more extensive reuse.

Reply: The reviewers' point, that is the leaching of the Zn component, is right. This was confirmed by the ICP-analysis of RCF reaction solution with fresh RuN/ZnO/C, in which Zn element was detected. Of note, a slowdown of Zn leaching was observed in further cycles, as seen in the cases of fresh, 1st recycled, and 2nd recycled (after calcinated regeneration) catalysts, the Zn contents were determined as 6.2, 4.9, and 4.5 wt%, respectively. In our previous work, we have reported that the calcination of spent catalyst can suppress further leaching of Zn during lignin hydrogenolysis (ChemSusChem, 2018, 11, 2114.).

Catalytic results indicated that the partial loss of Zn did not influence catalytic performance dramatically in birch RCF. The soluble Zn might not promote the lignin hydrogenolysis, because the combination of RuN/C and $\text{Zn}(\text{OAc})_2$ did not show higher performance than RuN/C in terms of activity and selectivity.

As suggested, we used MOF-derived Ru catalyst (Ru/ZnO/C-MOF) to treat birch under RCF conditions, which gave ca. 35 wt% yield of phenolic monomers with propenyl side-chain as major products. The hydrothermally treated Ru/ZnO/C-MOF catalyst showed a slight drop in monomers yields (ca. 30%). By comparison, H.T.-RuN/ZnO/C (reported in this manuscript) showed a consistent performance with fresh one. In this context, the coordinated N species in RuN/ZnO/C can enhance the stability of Ru centers.

These results have been added in the manuscript. We hope our responses will satisfy you.

As noted, the study is very well done and very extensive and would provide a very interesting addition to the literature on lignin first; impact and novelty wise a stronger case needs to be made in my opinion to be suitable for publication in Nature Commun.

Please find some additional comments below:

- While not part of course of the assessment of the merits of the paper, the manuscript is in need of a very careful and extensive language check.

Reply: Thanks for pointing out these problems. We have carefully checked the manuscript and English language has been edited by Author Services of Springer Nature.

- The authors should more explicitly compare their results with (own) literature; this holds for the catalyst synthesis part, but also for the use of a Lewis acid cocatalyst (cf. the system by Abu-Omar which contains Pd+Zn).

Reply: As suggested, we tested the catalytic performance of the combination of RuN/C and Zn(OAc)₂ (Abu-Omar's system) in RCF of birch, by which phenolic monomers were obtained in 21.4 wt% yield. This result is similar to that from RuN/C (24.9 wt%), and lower than that from RuN/ZnO/C (46.4 wt%), suggesting that the anchored ZnO species serve an auxiliary role in lignin hydrogenolysis. These results and discussions have added in the manuscript and Supplementary Table 4.

- It would be interesting to provide more information on the dimer/trimer/oligomer components of the lignin oil obtained; the group of Sels have recently reported extensive analyses that could of use here.

Reply: In GPC profiles, two major signals ca. 183 and 428 g mol⁻¹ were ascribed to phenolic monomers and dimers, and a small broad peak corresponding to oligomers was also observed.

To analyze the dimers, the lignin oils were silylated and characterized on GC-MS. Supported by literatures, a total of 11 different dimers were identified, among which most had C-C linkages, except for a small number of 4-O-5 dimer.

These results have been added in the manuscript and Supplementary Information. Thank you for your professional suggestion.

Reviewer #2 (Remarks to the Author):

The authors present a new Ru catalyst for reductive catalytic fractionation of lignocellulosic materials. The results are fascinating and provide further insight into the relationships between catalyst and lignin's active stabilisation. The experimental evidence supports the claims. However, it is unclear how the authors could separate the catalyst from the pulp (on page 15, "To evaluate the reusability of RuN/ZnO/C, the spent catalyst isolated from the carbohydrate pulp was submitted for birch RCF directly"). The experimental details need to be presented in Experimental since catalyst recovery in RCF is still very problematic.

In summary, it is an exciting manuscript with very nice results.

Reply: Thanks very much for this important suggestion.

After RCF reaction, the solid phase containing carbohydrate pulp and catalyst was dried at room temperature, which was then transferred to a 100-mesh screening. The spent catalyst could be readily separated from the carbohydrate pulp by sieving, because the carbohydrate remained the original framework of biomass (2-5 mm) without collapse. The detailed scheme and procedure for the separation of catalyst and pulp have been added as Supplementary Fig. 25 (as shown below).

Reviewer #3 (Remarks to the Author):

The authors present a nice study on developing a Ru single atom catalyst for reductive catalytic fractionation of biomass. The manuscript needs to be edited for minor grammatical mistakes (e.g., articles). The manuscript handles the synthesis of a new catalyst, detailed characterization of the catalyst, the activity of the catalyst in the hydrogenolysis of birch sawdust, effect of reaction conditions on hydrogenolysis yield, effect of solvent and comparison of different biomass sources. Authors put effort on completing the picture of several aspects of the new catalyst and compared it to standards catalysts. It is a very dense manuscript, however there are some points need to be clarified. The manuscript could be considered for Nature Communications after addressing the issues listed below:

1. For the determination of the biomass composition, NREL procedure was used. The total composition obtained was found to be 87.2%. This looks a bit low for total composition analysis. Generally, it would be 95+% by the NREL protocol. Why this is so low? Did you grind the biomass to the conditions?

Reply: Thank you for your kind comment. The biomass sawdust has been ground and screened through 60-mush sifter before NREL determination. If extracted components (3.5 wt%) was added, the total composition of birch would reach 90.7 wt%, albeit without considering water and ash. After comparison, current total composition obtained from birch is comparable to those reported in literatures (as shown below).

Samples	AIL (wt%)	ASL (wt%)	Cellulose (wt%)	Hemicellulose (wt%)	Extraction (wt%)	Ash (wt%)	Water (wt%)	Sum (wt%)	Ref.
Birch	19.1	-	39.3	20.7	2.5	-	-	81.6	EES, 2015, 8 1748
Birch	19.5	-	39.3	20.7	2.5	0.34	5	87.3	Science 2020 367,1358
Birch	23 (AIL+ASL)		38	25	3	-	-	89	Green Chem. 2017, 5767.
F-birch	19.7	2.6	37.3	22.5	3.6	0.6	-	86.3	ChemSusChem 2016, 9, 3280
Birch	19	-	44	21.3	3.2	-	8.7	96.2	Nature Cata. 2018, 1, 772
Birch	20.4 (AIL+ASL)		48.7	25.7	-	-	-	94.8	Nature Common. 2016, 7, 11162
Birch	22.7	2.6	42.3	19.6	3.5	-	-	90.7	This work

2. Is the catalyst pre-reduced before the reaction? If not, the authors should comment on the catalyst oxidation state under reaction conditions. One would expect methanol reforms and reduces the catalyst; however, catalyst characterization does not clarify this. Authors should comment of this important aspect of the catalyst.

Reply: please see the following response.

3. Related with the question 2, do you have any EXAFS study related with the reduction of Ru atoms? Did you perform the experiments before and after reduction and see the differences?

Reply to questions 2 and 3:

(1) In the manuscript, the RuN/ZnO/C catalyst was used directly without pre-reduction. As suggested, we treated RuN/ZnO/C in MeOH at 240 °C, H₂ (3 MPa at R.T.) for 4 h, which gave RuN/ZnO/C-(R) catalyst. In RCF of birch, RuN/ZnO/C-(R) can produce phenolic monomers in 46.1 wt% yield with 83% selectivity to **Pr-G** and **Pr-S**, being in line with RuN/ZnO/C.

(2) To realize the oxidation state of Ru in RCF reactions, we performed XAFS measurements for RuN/ZnO/C-(R) (see below). The Ru K-edge XANES spectra showed that the near-edge of RuN/ZnO/C-(R) was located between Ru foil and RuO₂, revealing the positively charged state of Ru. The EXAFS spectrum of the Ru K-edge in R space displayed a primary peak situated at 2.3 Å, corresponding to Ru–Ru coordination. These results were in accord with the scenarios in HAADF-STEM images, where both atomically dispersed Ru single atoms and Ru nanoclusters were detected (Supplementary Fig. 5). Obviously, most of the single Ru atoms were reduced into nanoclusters during pre-reduction process.

(3) Since there was no difference of catalytic performance between RuN/ZnO/C and RuN/ZnO/C-(R), Ru nanoclusters in RuN/ZnO/C-(R) should be the active species for lignin hydrogenolysis. The as-formed Ru nanoclusters should be beneficial to the dissociation of H₂, an uphill process in lignin hydrogenolysis, because H₂ dissociation on nanoparticles *via* homolytic path is more convenient than that on single atoms *via* heterolytic path.

Given XAFS measurements of RuN/ZnO/C and RuN/ZnO/C-(R) were respectively performed at Shanghai Synchrotron Radiation Facility (SSRF) and Beijing Synchrotron Radiation Facility (BSRF) (SSRF was closed recently due to COVID-19), we did not combine the XANES spectra.

We really appreciate this reviewer for this forward-looking question, which offered insight into understanding the possible pathway of lignin hydrogenolysis under such a catalyst. These results and discussion have been added in the manuscript.

Ru K-edge XANES and FT-EXAFS spectra of RuN/ZnO/C-(R)

4. Regarding Figure 4c: Liquid phase reactions are expected not to be affected by the pressure change. At the pressures studied, majority of the reaction mixture is in the gas phase. The authors should provide thermodynamic calculations for the conditions used. Also, are these the final pressures or the set pressures at the room temperature?

Reply: The H₂ pressure showed in previous manuscript was the set pressure at room temperature, which would be raised with the rise of temperature. Regarding Figure 4c, H₂ pressure was set to 0.1 to 4 MPa at room temperature, and the final pressures increased to 5.0, 6.6, 8.6 and 10.0 MPa at 220 °C, respectively. Thermodynamic calculations indicated that MeOH under such conditions remained as liquid phase, except 0.1 MPa (R.T.) H₂ (see Supplementary Table 8).

We calculated the mole fraction solubility of H₂ in MeOH based on a reported model, which displayed a linear relationship with the total monomers yields from RCF of birch (Supplementary Fig. 17, see also below). This indicated that sufficient H₂ transfer from gas to liquid phase is necessary for current lignin hydrogenolysis.

We are grateful for this reviewer for pointing out these issues. These results and discussions have been added in the manuscript and Supplementary Information.

5. On page 22, the procedure for the time profile experiment is described. It is specified that the reaction pressure was kept at 3 MPa at 200 °C. According to the phase diagram, under these conditions, methanol is expected to be completely in gas phase. Is 3 MPa your operating pressure? If so, this complicates the reaction system, and may affect the reliability of the results.

Reply: Thanks for pointing out this mistake. 3 MPa is the set H₂ pressure at the room temperature, and the final pressure was kept at 7.2 MPa (this reaction was performed in a 300 mL reactor with a constant pressure complementary H₂ device). Phase diagram suggested that MeOH kept as a liquid phase under such a condition. This mistake has been revised in the manuscript.

6. Supplementary Information, page 4 describes the analytical procedure for calculating the monomer yield. In the first paragraph, instead of "...available standards or independent synthesis.", "...available standards or independently synthesized standards." would be easier to understand.

Reply: The description of "...available standards or independent synthesis." has been revised as "...available standards or independently synthesized standards."

7. In the formula given in supplementary information, on page 4, what does M (total monomers) correspond to? Is this the mass of the detected and quantified monomers (e.g. Pr-G, Pe-G, POH-S,...) or is it referring to the mass after converting the moles of quantified monomers to the moles of monolignols (coniferyl alcohol, sinapyl alcohol) and their corresponding mass? The equation should be referring to the mass of the monolignols found in the biomass to determine the weight percent of Klason lignin.

Reply: M (Total monomers) refers to the mass of the quantified total monomers, and the description has been modified as "Mass (Total monomers)" in the Supplementary Information.

8. Could authors provide the details of GC/MS method (temperature, ramp, calibration range)?

Reply: The following GC or GC-MS procedure was used: 1 μL aliquot with a split ratio of 20:1, injection temperature of 250 $^{\circ}\text{C}$, column temperature program: 50 $^{\circ}\text{C}$ (hold time 3 min), 8 $^{\circ}\text{C min}^{-1}$ to 280 $^{\circ}\text{C}$ (hold time 5 min), detection temperature of 290 $^{\circ}\text{C}$ (for FID) or 280 $^{\circ}\text{C}$ (for MS). The quantification of monomers in the lignin oily samples were evaluated with authentic samples by using commercially available standards or independently synthesized standards. The as-obtained lignin oils were diluted to 10 mg/mL in ethyl acetate containing internal standard (tetradecane) before GC and GC-MS systems. Linear calibration curves were produced from 0.14 mg/mL to 3.9 mg/mL for authentic samples.

The details of GC/MS method have been added in the Supplementary Information.

9. On the supplementary information, page 5, what is the calibrated M_w range? Was a calibration kit used? If so, what is the part number and supplier?

Reply: The standard curve was made by polystyrene standards having different molecular weights, including 162 g mol^{-1} , 370 g mol^{-1} , 580 g mol^{-1} , 860 g mol^{-1} , and 1320 g mol^{-1} . The polystyrene standards were purchased from Agilent Technologies (polystyrene calibration kit S-L2-10, Part Number PL 2010-0105). The detailed information on polystyrene standards has been described in the Supplementary Information.

10. Table 1 of supplementary information lists the ICP-AES results of the fresh and spent catalyst. It looks like some of the Zn leaches. Was that confirmed by analyzing the liquid mixture? The authors should comment on the effect of Zn leaching on catalyst stability.

Reply: Thanks for pointing out this. As suggested, we measured the RCF reaction solution (obtained from fresh RuN/ZnO/C) by ICP-analysis, in which Zn element was detected. This result confirmed that some of the Zn has leached. We also noticed that Zn leaching become slow in further cycles, as seen in the cases of fresh, 2nd use, and 3rd use (after calcinated regeneration) catalysts, the Zn contents were determined as 6.2, 4.9, and 4.5 wt%, respectively. Catalytic results

indicated that the partial loss of Zn did not influence catalytic performance dramatically in birch RCF.

11. On page 19, line 458, Fig. 7 should be Fig. 8.

Reply: Fig.7 in Page 19 has been revised as Fig. 8.

12. The references are not in the same format.

Reply: We rechecked and modified the format of the references. I am truly grateful for your kindness.

In conclusion, I believe this work might be publishable but only after major revisions including additional experimental data.

REVIEWERS' COMMENTS

Reviewer #1 (Remarks to the Author):

The authors have addressed the comments and questions put forward by the reviewers, providing a considerable amount of new data, including information on Ru oxidation state in the catalyst, the nature of the dimers in the lignin oil and more data on the stability of the catalyst at hand (and the MOF-derived one for comparison). The additional data provides a valuable addition and has further improved the already extensive and detailed manuscript. The Ru oxidation state and coordination data from the XANES and EXAFS shows the expected reduction upon treatment in MeOH (and the catalytic tests show this reduced catalyst to perform equally well), but also shows clustering of Ru (as evidenced by the Ru-Ru nearest neighbor distances seen in the EXAFS). The authors then conclude that it is 'obviously' the Ru clusters that are the catalytically active species. This may indeed be the case (but it is not necessarily so; some redistribution can still occur under reactive conditions and the dispersion/speciation of Ru may differ when reduced in the presence of the woody biomass in MeOH) and if so this does change the narrative a bit, as first atomic dispersion was assumed to hold and this is (still) reflected throughout the text (from the title onwards). The new observation should therefore more consistently be incorporated throughout the manuscript; i.e. the synthesis yields a mono-atomic catalyst precursor and that this one, likely, evolves into a more complex, but still highly disperse catalyst for the reaction. The manuscript seems to be of two minds currently and this should be addressed (in the title, introduction, and in the results/discussion sections).

Second, the authors comment at length in the rebuttal and in the text on the stability of the catalyst and rate this as 'excellent' on several occasions. While good reusability is demonstrated, the loss of Zn from the catalyst and the modest drop in activity/selectivity seen does not fully warrant such an adjective; I would suggest a more modest description.

Finally, the manuscript can still do with a careful (language) check in my opinion (e.g. the first and last sentences of the abstract are somewhat odd; page2line53 'initially enabled' needs to be rephrased; p9|238 'to realize' -> 'to study'; p13|329 'was still sidelined' should be rephrased; figure 6 'directed' -> 'direct reuse'; p17|423 'dominat');).

If the above is taken into account, in particular the single atom/cluster comment, the paper can be accepted for publication in Nature Communications.

Reviewer #3 (Remarks to the Author):

The authors have adequately addressed all my concerns and comments. I would commend the authors for careful and detailed responses to all of my questions. I can now recommend the publication of the manuscript in Nature Comms.

The point-by-point response to the reviewers' comments

Reviewer #1 (Remarks to the Author):

The authors have addressed the comments and questions put forward by the reviewers, providing a considerable amount of new data, including information on Ru oxidation state in the catalyst, the nature of the dimers in the lignin oil and more data on the stability of the catalyst at hand (and the MOF-derived one for comparison). The additional data provides a valuable addition and has further improved the already extensive and detailed manuscript. The Ru oxidation state and coordination data from the XANES and EXAFS shows the expected reduction upon treatment in MeOH (and the catalytic tests show this reduced catalyst to perform equally well), but also shows clustering of Ru (as evidenced by the Ru-Ru nearest neighbor distances seen in the EXAFS). The authors then conclude that it is 'obviously' the Ru clusters that are the catalytically active species. This may indeed be the case (but it is not necessarily so; some redistribution can still occur under reactive conditions and the dispersion/speciation of Ru may differ when reduced in the presence of the woody biomass in MeOH) and if so this does change the narrative a bit, as first atomic dispersion was assumed to hold and this is (still) reflected throughout the text (from the title onwards). The new observation should therefore more consistently be incorporated throughout the manuscript; i.e. the synthesis yields a mono-atomic catalyst precursor and that this one, likely, evolves into a more complex, but still highly disperse catalyst for the reaction. The manuscript seems to be of two minds currently and this should be addressed (in the title, introduction, and in the results/discussion sections).

Reply: Thank you for your professional comments. The result of "the transformation from single Ru atom to nanoclusters" has been discussed in introduction, results, and discussion sections. The role of Ru nanoclusters was described as "The as-formed Ru nanoclusters, probably acts as the catalytically active species for lignin hydrogenolysis". In this context, the description of "atomically dispersed ruthenium" in the title has been modified as "highly dispersed ruthenium".

Second, the authors comment at length in the rebuttal and in the text on the stability of the catalyst and rate this as 'excellent' on several occasions. While good reusability is demonstrated, the loss of Zn from the catalyst and the modest drop in activity/selectivity seen does not fully warrant such an adjective; I would suggest a more modest description.

Reply: As suggested, the description of "excellent stability" in abstract has been modified as "good stability".

Finally, the manuscript can still do with a careful (language) check in my opinion (e.g. the first and last sentences of the abstract are somewhat odd; page2line53 'initially enabled' needs to be rephrased; p9|238 'to realize' -> 'to study'; p13|329 'was still sidelined' should be rephrased; figure 6 'directed' -> 'direct reuse'; p17|423 'dominat');).

Reply: Thanks for pointing out these syntax errors. We have rechecked and revised the manuscript carefully.

If the above is taken into account, in particular the single atom/cluster comment, the paper can be accepted for publication in Nature Communications.